# Mentoring of oral health professionals is crucial to improving access to care for people with special needs

**Mathew Albert Wei Ting Lim**[1,2,3]*, **Sharon Andrea Corinne Liberali**[4,5], **Hanny Calache**[1,6], **Peter Parashos**[1], **Gelsomina Lucia Borromeo**[1]

1 Melbourne Dental School, University of Melbourne, Melbourne, Australia, 2 Dental Services, Alfred Health, Melbourne, Australia, 3 Dental and Maxillofacial Surgery Clinic, Royal Melbourne Hospital, Melbourne, Australia, 4 Special Needs Unit, Adelaide Dental Hospital, Adelaide, Australia, 5 Adelaide Dental School, University of Adelaide, Adelaide, Australia, 6 Deakin Health Economics, Institute for Health Transformation, Deakin University, Geelong, Australia

* mathew.lim@alfred.org.au

**Data Availability Statement:** All relevant data are within the paper.

**Funding:** The authors received no specific funding for this work.

## Abstract

### Introduction

Individuals with special health care needs continue to experience difficulties with accessing regular dental care. This has largely been due to clinicians feeling they lack the training and experience to manage their needs. The aim of this study was to determine whether working closely with specialists in special need dentistry influenced the willingness of clinicians to treat patients with special needs.

### Materials and methods

Semi-structured interviews were conducted with specialists and clinicians involved in these mentoring initiatives. Qualitative thematic analysis was used to determine perspectives towards how this additional support influenced their willingness to treat individuals with special needs.

### Results

The views of all participants towards these supports were positive with clinicians feeling it not only offered them opportunities to learn from the specialists, but also increased their willingness to treat individuals with special needs and the timeliness and quality of care they were able to provide. Likewise, despite some concerns about the inappropriate use of specialist support, the specialists felt these mentoring relationships offered many benefits including improving timely access to care and ensuring individuals were able to receive appropriate care.

### Conclusions

Mentoring provided by specialists in special needs dentistry improved the willingness of clinicians to provide care for individuals with special health care needs. Supports such as

**Competing interests:** The authors have declared that no competing interests exist.

these are likely to be crucial to overcoming concerns of clinicians about their ability to manage the needs of these individuals and begin to address a significant barrier to access of care for individuals with special health care needs.

## Introduction

Oral health and access to regular dental care remain ongoing concerns for individuals with disabilities and complex health issues [1–3]. In many cases, these individuals are at higher risk of oral conditions, such as dental caries and periodontal disease, due to the confounding effects of their general health, polypharmacy, other impairments, and dependence on carers and family to assist with activities of daily living including oral hygiene care. These patient factors are further complicated by the lack of willingness of many general dentists to manage individuals with these more complex needs [2, 4].

A barrier for many oral health professionals is their perceived lack of knowledge or training in managing those with special needs [2, 4–6]. The ability to find suitably-trained and experienced dentists has been identified as a key barrier to accessing regular and ongoing oral health care by individuals with disabilities and their carers [2] and dental specialists in this field [7]. Oral health professionals have discussed their limited exposure to treating individuals with special needs as part of their training and difficulty with accessing ongoing professional development in this area as concerns [5, 6]. However, many have also suggested that receiving additional support may improve both their willingness to treat these individuals and reduce their reliance on referral to specialists where these patients often encounter long waiting lists [5, 6].

The dental and wider health literature have primarily focused on the need for competencies that reflect the growing necessity for health professional graduates to be trained in managing individuals with special health care needs [8–10]. However, improvements of this manner do little to address perceived deficiencies in training and experience amongst qualified health professionals. One support that oral health clinicians have suggested is the ability to work closely or alongside specialists at their local clinics, feeling that this may facilitate opportunities for mentorship, observation, consultation, and advice to overcome some of their own reservations about treating patients with special needs [5, 6].

Mentoring has become a vital part of addressing the complexities and demands of contemporary health care. In dentistry, mentoring programs are commonplace to help newly-graduated oral health professionals adjust from the formal education to clinical practice [11, 12], with it providing young clinicians opportunities to consult more senior colleagues about complex cases in addition to receiving psychosocial support with difficulties experienced. Although its use with more experienced clinicians in the dental setting is limited, it has been described within the academic environment as effective in developing skill sets crucial for clinical teaching and research [13, 14].

The use of mentoring is more commonplace in the medical and nursing fields. In particular, peer mentoring is embedded as a crucial part of junior doctor and nurse support in acquiring and refining new skills and competencies within a hospital or general practice setting [15–18] and for support mental health and resilience [19–21]. Likewise, as a result of the hierarchical nature of hospital environments, mentoring is often a component within specialty fellowship training in medicine [22–25], providing guidance in developing new knowledge and skills. Despite its limited use in the dental setting, potentially due to the less team-based approach in much of dental practice compared with other parts of healthcare, it seems apparent that many of the benefits seen in other health settings could be translatable and beneficial for oral health professionals.

Consequently, several dental clinics have explored mentoring as a supportive measure for clinicians and to reduce the necessity to refer to specialist services, particularly for managing patients with special healthcare needs, but no research has been completed to understand the potential impact of such programs on clinician perceptions and views. The aim of this study was to determine whether having a specialist in special needs dentistry work closely alongside other clinicians and provide opportunities for support and mentoring would affect the willingness of these clinicians to treat individuals with special needs. It was hypothesised that specialist mentoring would positively influence and improve the willingness of the clinicians receiving support to attempt treatment rather than immediately referring to specialist services.

## Materials and methods

Qualitative methods were used to explore perceptions of the support provided by specialists or clinicians with additional training in special needs dentistry to other clinicians in relation to managing individuals with heightened health care needs [26]. In particular, this study was focused on weekly clinical placements of specialists-in-training in special needs dentistry from the University of Melbourne at Carrington Health and Link Health and Community, two community dental clinics in Victoria, Australia, and the fly-in fly-out support and advice offered by a specialist in special needs dentistry to Top End Oral Health Services in the Northern Territory, Australia.

Consequently, a qualitative approach was determined to be most appropriate to understand the experiences of clinicians and to explore their perceptions and how these may have changed as a result of the additional support provided to them. In addition to being able to appreciate the individual experiences of participants, inductive thematic analysis of these views and the sentiments and themes that arose from the responses was used to develop an understanding of the collective response to these mentoring relationships.

Clinicians working at clinics with these support initiatives were invited to participate in a semi-structured interview to discuss their experiences of managing the oral health care of individuals with special needs. A promotional flyer was distributed to clinicians by the senior dentist or clinic manager and interested individuals asked to contact the research team. Specialists and specialists-in-training in special needs dentistry, dentists, oral health therapists, dental hygienists, dental therapists, and dental prosthetists who were either involved in the delivery of, or exposed to, these mentoring programs were eligible to participate.

The interviews were conducted in person or by phone or similar media, depending on the preference of the participants. A question guide was developed by the research team (Table 1) which used open-ended questions to determine the views of participants towards the

**Table 1. Question guides for interviews.**

**General dentists**
1. Can you tell me a bit about yourself and where you work?
2. Can you tell me about what groups of patients with special need you commonly treat at your clinic?
3. Have there been any additional supports provided to you to help you manage these patients?
4. Can you tell me a bit more about having the specialist / special needs postgraduate working with you?
 a. Can you describe how it works?
 b. What have been the benefits / shortcomings?
 c. Has it changed your perception towards treating patients with special needs?
5. If you could have access to anything else that might improve your ability to manage patients with special needs, what would it be?

**Specialists in Special Needs Dentistry**
1. Can you tell me a bit about yourself and your role?
2. Can you tell me about working alongside other clinicians at . . .?
 a. Do you feel it has been effective?
 b. Do you think it has improved the willingness of dentists to treat these patients?
3. Can you think of how this arrangement could be improved?

additional supports provided and if it impacted on the willingness or ability of clinicians to treat patients with special needs. Participants provided written informed consent for their involvement. All interviews were digitally-recorded (audio only) using two separate devices to ensure audio clarity. All interviews were conducted by a single member of the research team to ensure consistency in the interview process. Interview recordings were regularly audited by the interviewer and principal investigator of the project to ensure consistency.

Prior to undergoing analysis, the clearer digital recording of the interview was chosen and professionally transcribed. The transcription of the recording was then checked for accuracy against the recording with the second recording used to check in instances where further clarity of the response was required.

Analysis involved an initial reading of the responses prior to coding of emerging themes and sentiments using inductive thematic analysis and the qualitative data analysis software, NVivo (QSR International, Melbourne Vic, Australia) [26]. Initial themes that emerged from the first coding process were then reviewed and discussed by members of the research team. The research team had a range of backgrounds including specialists in special needs dentistry, specialists from other dental specialties, and a general dentist. Grouping of sentiments and themes that emerging from the views of participants of the study (Figs 1 and 2), and determination of data saturation based on the emergent themes, were thus the result of these discussions and consensus of the research team collectively.

Ethics approval was granted by the Melbourne Dental School Human Ethics Advisory Group (Ethics ID: 1544156.2) and the Human Research Ethics Committee of the Northern Territory Department of Health and Menzies School of Health Research (HREC Ref No. 2019–3364).

## Results

A total of 10 clinicians agreed to participate in this study. Three participants were classified as 'specialists' because of their involvement in providing the support to other clinicians. This

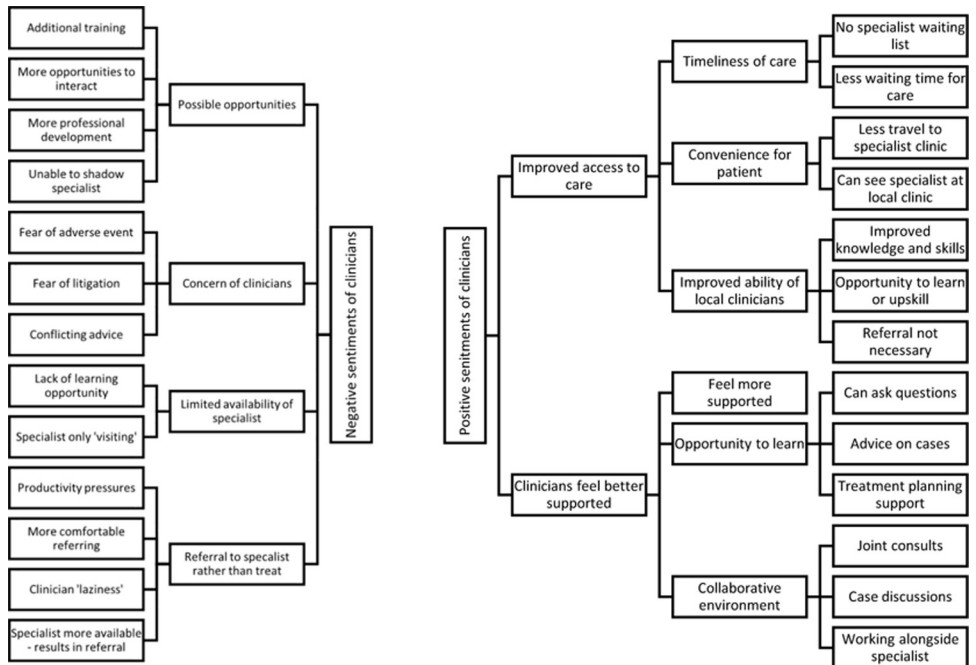

**Fig 1. Coding tree for clinician interviews.**

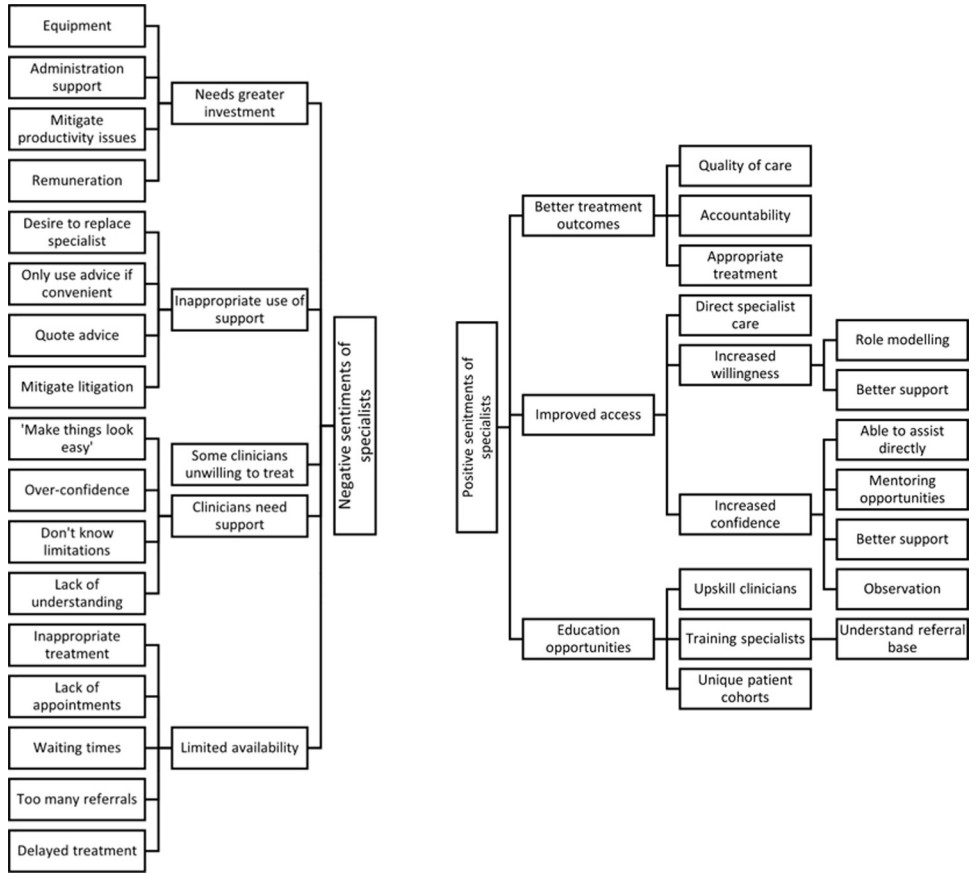

**Fig 2. Coding tree for specialist interviews.**

included one specialist in special needs dentistry and two training specialists in special needs dentistry. Remaining participants were 'clinicians' (6 dentists and 1 oral health therapist) with a range of experience (average years since graduation: 13.7, range: 3–23 years). Three of these clinicians had combined clinical and managerial roles (Clinicians 1–3).

### Responses of clinicians receiving support

Overall, the responses of clinicians receiving support from these initiatives were positive. Common themes that emerged were that the local availability of a specialist improved care for individuals with special needs in terms of both access and quality of care. For more complex cases, which were often referred to specialist services in locations where these were available, the additional support improved access to timely care by reducing the likelihood of external referral together with provision of treatment by the visiting specialist.

> *"The access for our patients to a higher level of care has been greatly enhanced because people can still come to our community clinic. They don't have to go to a specialist clinic, go on a massive waiting list, potentially travel far . . . They see clinicians who are at a higher level of knowledge and developing those skills."* (Clinician 1)

> *"I think the main advantage is that our patients don't have to travel into the city and the waiting time is much reduced. A lot of our patients are elderly or they need somebody to take*

*them, accompany them in for visits and so the location is very important. And the waiting time as well. If you were to send them in to the dental hospital there would be a longer waiting time."* (Clinician 5)

In addition, the ability of clinicians to draw on support from the specialist through these mentoring relationships improved their willingness and confidence in managing individuals with special needs. Supports extended from seeking advice, joint consults, and reviewing treatment completed by the specialist. As a result, clinicians often did not necessarily feel the need to refer to the specialist for treatment because it gave them the opportunity to learn from cases managed locally by specialists and they felt more supported to manage similar cases in the future.

*"Certainly having somebody around to take care of those complex patients is very helpful. And I also learnt reading through their dental records and seeing how they manage their patients . . . and if there's any questions I can email them and get their response."* (Clinician 4)

*"Well there was one case where it was quite helpful because I just booked that patient in for a consult with a special needs postgrad. I said I'm not really sure how to proceed. We had a few other medical issues going on and the postgrad was able to give me some guidance. . . that helped me crystallise my treatment plan to be able to provide that for the patients. So in that one case, the patient was not referred to the dental hospital after I sought some guidance from the special needs postgrad. . . I think it may help in the future a little bit in that I sort of know how to approach it."* (Clinician 5)

*"I found that a lot of the dentists often took up the opportunities to discuss with the special needs dentist and to get recommendations . . . about how to better manage patients rather than just referring off . . . it definitely was a more collaborative environment"* (Clinician 6)

*"Or if the case is particularly more difficult . . . we can postpone it until he comes in and then he will take on the case himself which has been very useful as well."* (Clinician 7)

The main drawbacks reported largely related to the availability of specialist support. This also differed in nature between the supports provided by the training specialists in Victoria versus the fly-in fly-out specialist in the Northern Territory due to the difference in frequency at which local support was provided for clinicians. In Victoria, where support was available on a weekly basis, the most commonly reported issue was that the lack of availability of specialists resulting in some clinicians deciding to refer patients rather than using the support to attempt management themselves. Clinicians felt that this reflected the unwillingness of some of the other clinicians to treat individuals with special needs or insufficient support or other constraints, such as productivity targets, that influenced their decisions.

*"The good aspect is that I believe my patients are getting the best care they are entitled to . . . The bad . . .it makes me very lazy. So I see a person who is specialising and . . . I tend not to do so much to push myself to read more into it or try to manage those cases. And so usually I get somebody to look after the patient for me."* (Clinician 4)

*"I think, most of the time, if there was an issue, to be honest, these patients would be referred to the postgrad themselves. I don't know how much of a learning experience we actually had . . . it's an easy way for some of the dentists to pass the buck."* (Clinician 5)

*"... certain clinicians who would probably just feel more comfortable referring... to the specialist ... I suppose it just comes down to, it's a little bit easier... And to put it very bluntly, right now within the public system... when your performance is just showing in your DWAUs* (Dentist Weighted Activity Units) *and you're told that the clinic may not have enough money to fund the current staffing levels if the DWAU units are not upheld, all of this liaising ... doesn't count toward your performance ... I'd rather see a special needs patient if I knew it would automatically attract a certain amount of DWAUs. I think it'd take some of the pressure off."* (Clinician 5)

In contrast, specialist support provided to clinicians in the Northern Territory was less regular. The specialist would fly in every few months to see patients with advice provided remotely on an as-need basis. As a consequence, the weaknesses that were identified were largely related to the difficulty of relying on advice to make clinical decisions and the limited opportunity to learn directly from the visiting specialist.

*"The trouble is ... you are sandwiched between the medical professional and the dental professional. Sometimes the dental professional ... may not accept what the medical professionals say. Then it puts me, or clinicians like me, under the light ... like any type of audit process."* (Clinician 3)

*"I think the main downside is that because (the specialist) only really visits, it's once every so often ... I won't be within the same clinic when he's working because we've only got a one chair clinic ... he's a busy person. He can't be here all the time. But when he is here, I wish I could either be working alongside him or, at least, see the way he does things. And so that we can work together a bit more."* (Clinician 7)

The potential to use the specialist as a learning resource and, in particular, the possibility of shadowing the specialist to improve their understanding of how specialists manage such cases, was a promising opportunity raised by many clinicians across all sites. Many felt this was likely to be the key to improving confidence in managing this group of patients into the future. However, a concern was that this support was largely limited by the availability of specialists.

*"I'd love for the postgrads to have more scope to provide training ... Because when they're here, they're going to be focused on their patients."* (Clinician 1)

*"(the specialist) might say he can only come once a month. But the patients we are seeing may be needing something anywhere between these times ... Increasing the frequency, that's one answer."* (Clinician 3)

*"To see how they deal with the patients and the different clinical skill sets that might bring. And also how they liaise with the different medicos. I really think there's no better way until I see how it's done. That might make it a bit easier for me to do it next time and to ask them questions in real time when they're going through a report that the oncologist faxes through."* (Clinician 5)

Despite these limitations, managers felt that the mentoring relationship had improved the ability of their clinicians to manage individuals with special needs and that it had been vital in upskilling their local workforce and for improving access to care.

*"The special need postgrads have helped develop their confidence and we're seeing that because one of the things with the overflow of the cancer clinic is, of course, not all the patients*

*can be seen by the postgrads. So there have been a number of clinicians who have started to gain a bit more experience in those cases and as they develop their confidence."* (Clinician 1)

*"I think having the special needs dentists there actually gives us the opportunity to discuss these cases . . . I can see that some clinicians are getting more comfortable and willing just to treat these patients . . . it's a great benefit for our clinic and could help us to see a few more of these special needs patients."* (Clinician 2)

### Responses of specialists providing support

The responses of specialists were generally positive and complemented those of the clinicians they mentored. The main strengths of the support provided included improved accountability of the clinicians and improved access to timely care. This was either the result of them providing direct care at the community level or an increased willingness of clinicians to manage patients with special needs stemming from their interactions.

*"I think probably the overarching impact I have had is one of quality. . .maintaining these clinicians' accountability to providing an appropriateness of healthcare to these clients." (Specialist 1)*

*"I was seeing one of the oncology patients in my room and another dentist was seeing a very similar case . . . I could help the dentist do the treatment plan for their patient . . . she was able to treat that [case] and said that she felt she may able to see some similar patients as a result . . . you can give the other dentist a little bit of understanding on how we do things . . . [and] some of the dentists, not all of them, feel more happy to do what we do now."* (Specialist 2)

*"I think for dentists, there was also an element of the special needs postgrad acting as a role model . . . they had the opportunity to see, in a sense, how the special needs postgrads managed particular cases . . . even if it was not by direct observation . . . And I think in some ways it probably would have also given them the opportunity to see that special needs patients could be managed in the community setting."* (Specialist 3)

The primary concern of specialists was that their limited availability impacted on patient care, and more so for the fly-in fly-out specialist service due to the relative irregularity of the service. This was largely related to their availability to provide or guide patient care. In their responses, specialists were able to provide examples of their reservations about clinicians understanding the needs of individuals with special needs.

*"[The clinic] would keep booking and booking patients. And sometimes . . . They need treatment before they started their oncology treatment. And I use to get them two to three months after they had started their treatment."* (Specialist 2)

*"The other issue is in terms of the services is that patients have to wait . . . the frequency of my travel to the Northern Territory certainly impacts on acute management of clients with special needs or those acute medical things."* (Specialist 1)

*"And then also the other thing is with my management of those with disability. If they have an acute dental condition then they are at the mercy of the general dentists. An example of that was a client with Down syndrome . . . she's got very few permanent teeth and we've been nursing along these deciduous teeth . . . over like a period of 20–30 years. She came in with an acute infection of one of her upper incisors and the general dentist took out all four and held her down while she was screaming under LA. This is somebody who's non-verbal with a severe*

*intellectual disability. I've had to do treatment before and it was completely fine. Whereas, you know, you had somebody who didn't have an understanding of the client. I wasn't there to protect her from this dental treatment . . . So this is a real risk of me not had been having frequent visits."* (Specialist 1)

As a result, specialists felt that the inherent weakness of any support provided was the need for clinicians to recognise the possible limitations of their own knowledge and scope of practice. Their sentiments suggested that clinicians did not have a full appreciation of the complexity of the patients they may be managing.

*"I think there's a lot of dentists that whilst they have part of the picture, part of the jigsaw puzzle, they don't have that overarching understanding of special needs management . . . In terms of special needs management, they're treating one aspect of their medical history and one risk factor without understanding the interplay between comorbidities and the special needs individual."* (Specialist 1)

*"The main disadvantage from my point of view is when they see. . . that we do things easier. The way we do things is difficult but when we do it over and over I know it looks easier even if it is not. So some other people may believe 'okay, this is so much easier than we thought. We may not need a specialist. We can do it because if they do it in such a way then we can do it'."* (Specialist 2)

Furthermore, specialists raised concerns that clinicians may have been using the support provided inappropriately. Rather than using it to improve their ability to manage individuals with special needs, there were suspicions that the specialist was used to mitigate medicolegal risk in difficult clinical situations or reduce the need to spend additional time with patients because of productivity pressures.

*"There are major issues, medicolegal issues . . . about providing advice for clients that I haven't examined . . . where you have a dentist who knows that they are going into a difficult high risk situation who is using me or (another specialist) to be able to put down in their notes 'as per the opinion of this specialist' . . . So that's purely using our names as a risk mitigation."* (Specialist 1)

*"In some cases the dentists were more willing to try because they could get the advice that they needed quite easily whereas in other cases, if a patient was going to take more time and impact on productivity and efficiency, I felt that I sometimes referred on for those reasons."* (Specialist 3)

Given these concerns, specialists felt that in order to fully support clinicians to the degree necessary, greater investment into the speciality was required. Their recommendations included appropriate remuneration for specialist support, increased time for specialist involvement, greater clerical support, and investment into facilities and equipment to assist all clinicians. Specialists felt that some of these measures would not only improve the care they could provide to patients, but also improve the willingness of other clinicians to manage these individuals or consider specialisation.

*"Given that neither of us are actually remunerated or contracted for that added support . . . It's an organisation not supporting the specialty of special needs dentistry but asking for it at the same time."* (Specialist 1)

*"Originally it was funded for eleven weeks a year specialist involvement time to now four . . . I would go back to having monthly visits and maybe sharing that between two specialists. But it would have to go back to what it was. You couldn't share the portion that I am on at the moment without just over complicating things and creating more issues."* (Specialist 1)

*"I would like to have two specialists or two postgrads . . . we could try and deliver the dental treatment faster instead of delaying the treatment because we can only see the patient say every Monday for two to three months. So, my plan will be to have two to three weeks to finish the treatment but . . . we need more people working in the clinic."* (Specialist 2)

*"I would say that one thing that would be good to have at that level would probably be more administration support because we were having to do a lot of chasing of medical history information. Having to do our own paperwork."* (Specialist 3)

*"It's not always practical but putting in a hoist or some sort of wheelchair lift would certainly help with our patients who had physical disabilities, but it is a large investment. And it may also require modification to the premises which may not be possible."* (Specialist 3)

Furthermore, specialists saw significant value in using training specialists to provide support to clinicians in such initiatives. Reflecting on the clinical placement of training specialists, there were suggestions that this mentoring could offer valuable opportunities in other settings and for learning for both training specialists and the clinicians they may be supporting.

*"It would be nice to get to use this resource because I guess there's a uniqueness to the Northern Territory and to that group. You've got the whole cultural overlay with the comorbidities. So I think it would be . . . if we were to have it as a place of training as well"* (Specialist 1)

*"(The placement) helps develop an appreciation of the challenges faced by general dentists with delivering care at the community level . . . if they haven't provided treatment in that space before."* (Specialist 3)

Regardless, specialists felt that while there may be some clinicians who would continue to be unwilling to treat individuals with special needs, there was a role for the specialty and employers to encourage and support clinicians in this area, particularly amongst those who do have a desire to assist in improving access to care for these individuals.

*"I think most general dentists are frightened of special needs clients, particularly those with complex medical issues, and I think they're more than happy for you to hold their hands in terms of guiding them through stuff and having a conversation with them because it's not necessarily just about them presenting a clinical scenario to you and you providing the answer, it's about actually having the conversation. And that's why the relationship between the specialist and the general practitioner is the most important thing in terms of the effort and how effective it is . . . (but) how effective I am is purely dependent on the relationship I have with the other clinicians . . . [I'm] far more willing and able to support dentists who I believe are working with me."* (Specialist 1)

*"It also depends on the dentist willingness to treat this kind of patient . . . I've been in places where I can see dentists are happy to treat these patients but at the same time there is a portion of dentists that just basically refuse in a very polite way . . . one of the main things they need are dentists who can treat patients with special needs. So there shouldn't be any easy way out because this is one of their requirements of being able to work in the public sector."* (Specialist 2)

*"On the one hand, potentially the patients would be seeing somebody who has better skills and confidence in managing their case (if there was more specialist involvement). On the other hand, it decreases the opportunity for the general dentist to manage those cases and decreases their confidence because from experience . . . you just have to deal with those cases. And you know, in that way you gain the experience to manage it routinely yourself. So that's really a double edge sword."* (Specialist 3)

*"So, one example I can think of is . . . at a clinic that I personally haven't worked at but I know other postgrads worked at. And after, the [postgrad] program . . . had to be suspended at that site, that particular clinic was able to build on some of the work that had been started . . . . I spoke to one of the dentists who had been there and they felt that they were more confident about communicating with the hospitals after the postgraduates had left. So I think that's pretty positive that the program will continue." (Specialist 3)*

## Discussion

Improving the oral health of individuals with disabilities and complex medical issues needs to be considered as a high priority. Unfortunately, because other medical issues have often taken precedence over oral health, preventable dental conditions and oral health are neglected resulting in undue suffering from dental pain [27]. Furthermore, when these individuals finally seek dental care, their treatment needs are often greater, more complex, and may be more difficult to manage. The resultant delay in receiving this treatment can often result in unnecessary pain and tooth loss, creating a cycle that results in further debilitation and compromised masticatory function, swallowing, speech, and aesthetics [27].

Whilst part of the issue is a need for oral health to be given the same priority as general health, there is also need for concerted efforts from the health care system to ensure all patients are able to receive timely care. In the dental profession, this continues to be a problem with individuals with special needs reporting difficulties identifying dentists willing to treat them because of a lack of perceived training, knowledge, and skills [2, 4]. Likewise, specialist services for those with special needs are primarily within the public dental system in Australia and often have long waiting lists due to an ever-increasing demand placed on a small specialist workforce and limited resources to provide these services [28, 29].

Dental services have taken different approaches to trying to address this issue with many providing additional support to their clinicians to try and reduce a reliance on specialist referral. A common theme from discussions with oral health professionals has been a desire to work closely or alongside specialists in special needs dentistry or for opportunities for mentoring [5, 6, 30]. This was the approach taken by the clinics involved in this study. In particular, two community dental clinics in Victoria, Australia (Link Health and Community and Carrington Health) collaborated with the Melbourne Dental School at the University of Melbourne to have training specialists in special needs dentistry work alongside dental teams consisting of general dentists, oral health therapists, and dental prosthetists once a week as a clinical placement. Similarly, the Top End Oral Health Services in the Northern Territory, Australia developed a formal relationship with a specialist in special needs dentistry from interstate to visit Darwin and Alice Springs several times a year to provide additional support to local clinicians. Between these periods, this specialist offered clinicians remote support by providing advice and input on individual cases based on the needs of individual practitioners.

The results of the current study indicated that despite some of the inherent differences between these two initiatives, overall they had a positive effect on clinicians working at each of the respective sites and improved their willingness to try to manage these patients rather than

referring them onwards. A key benefit was that clinicians were able to interact directly with the specialists in whatever manner reflected their own needs. Many participants and specialists discussed seeking advice on cases, observing or learning from specialists with respect to how they managed patients, the ability for joint consultations and treatment planning sessions, opportunities for role modelling, and aspects of moral support from the specialists in the possible situation of an adverse event.

A significant opportunity of working alongside specialists or specialist trainees, over other forms of interactions, was the ability for clinicians at the community clinic to experience direct mentoring and see how more-experienced clinicians worked in their local setting. From other initiatives discussed in the literature, a reported limitation has been a sense that specialists may not appreciate the challenges faced by clinicians at the community level because of limitations in local infrastructure, such as clinic design, equipment, and facilities with these commonly identified by oral health professionals as a barrier to treating individuals with special needs [5, 6, 30]. However, in this study, and particularly for those in Melbourne clinics working with training specialists, being able to see specialists work with them and with equal constraints allowed the local clinicians to see how they could adapt their own ways of managing patients; something more difficult to achieve through remote support.

In addition, the ability for these visiting specialists to improve access to care was essentially two-fold. Not only did their support improve the willingness of other clinicians to attempt treatment for these patients, but having the specialist work locally allowed patients to access specialist level care closer to home. Given that most specialist clinics in Australia are centralised in major cities, this may offer a significant advantage for many patients by reducing the burden of travel often placed on patients, families, and carers as a form of outreach or intermediate level service [2, 28].

Although many of the reflections of the general clinicians were somewhat anticipated, some interesting themes emerged from the views of specialists involved in providing this support. Although their reflections were generally positive and complementary towards many of the clinicians they supported, some discussed their role as one of ensuring quality and accountability. In doing so, they raised concerns about the ability of the clinicians they were supporting to appreciate the true complexity of cases and their management and discussed the need for them to intervene and ensure patients received an appropriate level of care. Previous studies into support provided by specialists in special needs dentistry have raised similar concerns about finding an appropriate balance between encouraging clinicians to attempt treatment when they may not fully appreciate the complexity, ensuring that these vulnerable patients are receiving the standard of care they deserve, and ensuring adequate access to this case [30].

In addition, specialist services often encounter the problem of being placed under additional pressure because of 'unnecessary' referrals or those that are perceived by specialists to lack complexity and could be managed by general dental practitioners [28]. The United Kingdom, in particular, has explored the use of case mix models to determine the complexity of patients treated by different clinicians and postulated its use for determining appropriate allocation of referrals to clinicians in specialist services [31, 32]. The problem with many such tools is they continue to rely on specialists, or those with additional training and experience, assessing the complexity of patients wherein the more significant problem lies with clinicians at the community level not being able to fully appreciate the interaction of social and medical factors in these patients. As a result, whilst tools that dentists may be able to use to assess the case complexity of patients with special needs are developed and validated, perhaps initiatives such as visiting specialists and mentoring, could also be used by specialist services to assist with assessment and triage of patients to determine the most appropriate provider for their care based on their needs.

Herein lies the dilemma faced by special needs dentistry as a specialty in many countries: how best can the needs of these patients be met without compromising the quality of care they receive? The solution is not increasing the reliance on specialist services, given the limited resources dedicated to these services and the limited specialist workforce. Instead, it is about ensuring that the dental workforce, and particularly those in the public dental system, are adequately equipped to manage the needs of this growing population.

The current study provides an example of how both the specialist and training specialist workforce may be used beyond direct patient care to encourage wider access to care. Whilst such initiatives may not necessarily change the perceptions of all clinicians and improve their willingness to manage this group of patients, it will go some way to addressing current deficiencies in training and experience that remain as significant barriers to access of care for these programs while training programs address these issues in future generations of oral health professionals.

Unfortunately, what is currently lacking is adequate resourcing which must include appropriate funding and priority dedicated to these patients, use of an integrated team of oral health professionals to manage preventive care for those more vulnerable to risk of oral disease, and ensuring this workforce has not only the adequate training and skills, but also appropriate investment into the specialty to benefit from ongoing support from specialists in this environment. In addition, efforts to improve the oral health awareness amongst other health professionals and carers in the disability and aged care communities is vital to supporting these preventive messages and increasing the priority of oral health in these populations.

There were several limitations of the current study. Firstly, due to the limited sample size, there was an inability to provide more in-depth comparison between the weekly support provided by the training specialists and the less regular support provided by the fly-in-fly-out specialist. Inherent differences between the support provided by the two models emerged from the responses of participants and certainly warrants further research of both models to allow health services to determine if such supports may be beneficial to their clinicians and availability of specialists. Furthermore, there was a risk of bias given that participants volunteered their time and views and that these may not have been reflective of the wider group of clinicians in these clinics. This reinforces the need for further research in this area to validate the initial results presenting in the current study.

Regardless of these issues, the results of the current study suggest that having more experienced clinicians, whether they are trainee or qualified specialists, to mentor other health professionals can have a positive influence and potentially improve the willingness of clinicians to treat individuals with special needs. Furthermore, these initial and promising results warrant further investigation into these and other supports that clinicians feel may improve their confidence, ability, or willingness to treat these patients. Although discussed primarily in the context of special needs dentistry, such models may have implications for other dental specialties in considering how specialists can continue to support oral health professionals, particularly in areas where access to specialists may be more difficult. Nevertheless, these findings reinforce the need for improved training of oral health professionals in the area of special needs dentistry and demonstrate the vital role specialists play in continuing to advocate for, and improve access to dental care and better oral health for individuals with special health care needs, not just in specialist clinics but also the wider community.

## Conclusions

The results of this study indicate that specialist mentoring improved the willingness of clinicians to treat patients with special needs. In particular, having opportunities to work alongside

each other often offered mutual benefits. From a specialist perspective, the support provided improved the quality of care received by patients at these clinics and ensured a greater level of accountability given the reliance of clinicians to recognise case complexity. In addition, where treatment was beyond the scope of clinicians, patients benefited from increased access to specialists treating them locally. Specialists raised concerns about possible inappropriate use of supports to mitigate medicolegal concerns and productivity pressure but for specialist trainees, it offered a unique opportunity to understand the barriers and challenges faced by clinicians they received referrals from. Likewise, working alongside specialists was positively received by clinicians and resulted in them feeling they were able to provide better access to care as well the opportunity to learn from specialists. Further research and investment is required into such initiatives to support clinicians working in the primary care setting to address existing barriers to accessing regular dental care experienced by individuals with special needs, particularly those related to the perceived lack of training and experience of oral health professionals in this area.

## Acknowledgments

Mathew Lim would like to acknowledge the support provided by the Australian Government Research Training Scholarship and the Rowden White Scholarship from the University of Melbourne.

## Author Contributions

**Conceptualization:** Mathew Albert Wei Ting Lim, Gelsomina Lucia Borromeo.

**Data curation:** Mathew Albert Wei Ting Lim.

**Formal analysis:** Mathew Albert Wei Ting Lim, Gelsomina Lucia Borromeo.

**Investigation:** Mathew Albert Wei Ting Lim, Gelsomina Lucia Borromeo.

**Methodology:** Mathew Albert Wei Ting Lim, Sharon Andrea Corinne Liberali, Gelsomina Lucia Borromeo.

**Project administration:** Mathew Albert Wei Ting Lim.

**Supervision:** Sharon Andrea Corinne Liberali, Hanny Calache, Peter Parashos, Gelsomina Lucia Borromeo.

**Writing – original draft:** Mathew Albert Wei Ting Lim.

**Writing – review & editing:** Mathew Albert Wei Ting Lim, Sharon Andrea Corinne Liberali, Hanny Calache, Peter Parashos, Gelsomina Lucia Borromeo.

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
