## [Decision Letter · Decision Letter 0]

10 Feb 2022

PONE-D-21-23289Mentoring of oral health professionals is crucial to improving access to care for people with special needsPLOS ONE

Dear Dr. Lim,

Thank you for submitting your manuscript to PLOS ONE. After careful consideration, we feel that it has merit but does not fully meet PLOS ONE’s publication criteria as it currently stands. Therefore, we invite you to submit a revised version of the manuscript that addresses the points raised during the review process.

We look forward to receiving your revised manuscript.

Kind regards,

Ratilal Lalloo

Academic Editor

PLOS ONE

Journal Requirements:

Additional Editor Comments:

Dear Authors;

The reviewers have suggested a number of minor revisions.

Reviewers' comments:

Reviewer's Responses to Questions

**Comments to the Author**

1. Is the manuscript technically sound, and do the data support the conclusions?

Reviewer #1: Yes

Reviewer #2: Yes

2. Has the statistical analysis been performed appropriately and rigorously? 

Reviewer #1: Yes

Reviewer #2: Yes

3. Have the authors made all data underlying the findings in their manuscript fully available?

Reviewer #1: Yes

Reviewer #2: Yes

4. Is the manuscript presented in an intelligible fashion and written in standard English?

Reviewer #1: Yes

Reviewer #2: Yes

5. Review Comments to the Author

Reviewer #1: This is overall a well written paper that uses qualitative research methodology to explore the impact of a model of care where the local dental workforce is upskilled by SND specialist input, and it nicely brings out the benefits of this.

The following minor revisions are suggested.

1. The authors should cite and discuss situations in other health disciplines where the approach described has been used with good effect, to bring out the general principles of the intervention, and its relevance to triaging according to case complexity. Examples from nursing and medicine would be informative.

2. Given differences in clinical scope for handling patients with medical complexity, was there a difference for the oral health therapist vs the general dentists with the approach used?

3. The references need to be formatted to be consistent, and to match PLoS One journal style. There are variations in abbreviations of journal names and in patterns of capitalization. The name of the journal in Ref 3 is given incorrectly.

4. Fig 1 could be classed as a table rather than a figure, perhaps, as it has no graphical elements.

5. In the Figure, the phrase "... what groups of patients with special need you commonly treat ..." should be

"...what groups of patients with special needs you commonly treat ..."

Reviewer #2: A well-articulated piece of qualitative research with clear data and rigorous analysis that support the conclusions of the impact and value of mentoring initiatives by SND specialists on the willingness of clinicians to provide care for individuals with special health care needs.

A key strength was the research team’s ability to demonstrate how the key findings fit within the existing literature. This was seen through the provision of rich contextual background in the introduction and discussion sections, thereby emphasising the clinical significance and translatability of research findings to the current sentiments of and challenges faced by clinicians and SND specialists on the ground. In addition, inductive thematic analyses were utilised appropriately that enabled the emergence of interesting themes. Nuances from the responses of participants were also skillfully identified, elaborated and analysed, giving rise to a thought-provoking and refreshing read.

Some minor points for the research team’s consideration would include elaboration on how data saturation was achieved, as well as provision of a coding tree to summarise the key emerging themes from the two unique groups of participants.

6. Reviewer #1: **Yes: **Laurence J Walsh

Reviewer #2: No

---

## [Author Response · Author response to Decision Letter 0]

19 Mar 2022

Dr Ratilal Lalloo

Saturday 19 March 2022

Dear Dr Lalloo,

Re: Revisions to manuscript – Mentoring of oral health professionals is crucial to improving access to care for people with special needs [PONE-D-21-23289]

Thank you for your correspondence and for the opportunity to revise our manuscript for submission to your journal.

Our research team have read and addressed the comments of the reviewers with our responses outlined below. As requested, we have uplodaded a marked-up and unmarked copy of the revised paper for your consideration.

Thank you for considering our manuscript for publication in your journal. We look forward to your response.

Kind regards,

Mathew Lim

Response to review comments:

Reviewer #1: This is overall a well written paper that uses qualitative research methodology to explore the impact of a model of care where the local dental workforce is upskilled by SND specialist input, and it nicely brings out the benefits of this.

The following minor revisions are suggested.

1. The authors should cite and discuss situations in other health disciplines where the approach described has been used with good effect, to bring out the general principles of the intervention, and its relevance to triaging according to case complexity. Examples from nursing and medicine would be informative.

We have added examples from the wider medical literature to the Introduction (P3 Lines 55-71) to demonstrate the benefit of mentoring and how the principles that underlie this practice may be applied to managing this patient cohort in the dental setting.

2. Given differences in clinical scope for handling patients with medical complexity, was there a difference for the oral health therapist vs the general dentists with the approach used?

Thank you for your interesting observation. Based on the responses of the specialists involved in our study, it was not apparent that a different approach was used to support general dentists and oral health therapists. Presumably, this was factored into the nature of advice provided but it did not specifically emerge from the discussions in this study and thus was not discussed in our manuscript. Likewise, given the relatively small number of OHTs that were working at these clinics and involved in treating these patients, it may not have been clear in this study, but would be an interesting consideration for future studies in this area.

3. The references need to be formatted to be consistent, and to match PLoS One journal style. There are variations in abbreviations of journal names and in patterns of capitalization. The name of the journal in Ref 3 is given incorrectly.

We have reviewed the reference list and made corrections to ensure consistency with the referencing style used.

4. Fig 1 could be classed as a table rather than a figure, perhaps, as it has no graphical elements.

As suggested, Figure 1 has been renamed Table 1.

5. In the Figure, the phrase "... what groups of patients with special need you commonly treat ..." should be

"...what groups of patients with special needs you commonly treat ..."

Thank you for identifying this typographical error. This has been corrected.

Reviewer #2: A well-articulated piece of qualitative research with clear data and rigorous analysis that support the conclusions of the impact and value of mentoring initiatives by SND specialists on the willingness of clinicians to provide care for individuals with special health care needs.

A key strength was the research team’s ability to demonstrate how the key findings fit within the existing literature. This was seen through the provision of rich contextual background in the introduction and discussion sections, thereby emphasising the clinical significance and translatability of research findings to the current sentiments of and challenges faced by clinicians and SND specialists on the ground. In addition, inductive thematic analyses were utilised appropriately that enabled the emergence of interesting themes. Nuances from the responses of participants were also skillfully identified, elaborated and analysed, giving rise to a thought-provoking and refreshing read.

Some minor points for the research team’s consideration would include elaboration on how data saturation was achieved, as well as provision of a coding tree to summarise the key emerging themes from the two unique groups of participants.

We have added further comment to the Methods section to clarify that data saturation was considered achieved by consensus of the research team on review of the coding progressively throughout data collection (P6 Line 131). Coding trees are provided for the clinician and specialist groups in Figures 1 and 2 for additional clarity.

---

## [Decision Letter · Decision Letter 1]

30 Mar 2022

Mentoring of oral health professionals is crucial to improving access to care for people with special needs

PONE-D-21-23289R1

Dear Dr. Lim,

We’re pleased to inform you that your manuscript has been judged scientifically suitable for publication and will be formally accepted for publication once it meets all outstanding technical requirements.

Kind regards,

Ratilal Lalloo

Academic Editor

PLOS ONE

**Comments to the Author**

1. If the authors have adequately addressed your comments raised in a previous round of review?

Reviewer #1: All comments have been addressed

Reviewer #2: All comments have been addressed

2. Is the manuscript technically sound, and do the data support the conclusions?

Reviewer #1: Yes

Reviewer #2: Yes

3. Has the statistical analysis been performed appropriately and rigorously? 

Reviewer #1: Yes

Reviewer #2: Yes

4. Have the authors made all data underlying the findings in their manuscript fully available?

Reviewer #1: Yes

Reviewer #2: Yes

5. Is the manuscript presented in an intelligible fashion and written in standard English?

Reviewer #1: Yes

Reviewer #2: Yes

6. Review Comments to the Author

Reviewer #1: (No Response)

Reviewer #2: Thank you for addressing the suggested comments and I look forward to the publication of this work.

7.

Reviewer #1: **Yes: **Laurence J Walsh

Reviewer #2: No

---

## [Editor Report · Acceptance letter]

13 Apr 2022

PONE-D-21-23289R1 

Mentoring of oral health professionals is crucial to improving access to care for people with special needs 

Dear Dr. Lim:

I'm pleased to inform you that your manuscript has been deemed suitable for publication in PLOS ONE. Congratulations! Your manuscript is now with our production department. 

Kind regards, 

on behalf of

Dr. Ratilal Lalloo 

Academic Editor

PLOS ONE